# Wounding triggers MIRO-1 dependent mitochondrial fragmentation that accelerates epidermal wound closure through oxidative signaling

Hongying Fu[1,6], Hengda Zhou[1,2,6], Xinghai Yu[3], Jingxiu Xu[1,2], Jinghua Zhou[1], Xinan Meng[1], Jianzhi Zhao[1], Yu Zhou [3], Andrew D. Chisholm[4] & Suhong Xu [1,2,5 ✉]

Organisms respond to tissue damage through the upregulation of protective responses which restore tissue structure and metabolic function. Mitochondria are key sources of intracellular oxidative metabolic signals that maintain cellular homeostasis. Here we report that tissue and cellular wounding triggers rapid and reversible mitochondrial fragmentation. Elevated mito-chondrial fragmentation either in *fzo-1* fusion-defective mutants or after acute drug treatment accelerates actin-based wound closure. Wounding triggered mitochondrial fragmentation is independent of the GTPase DRP-1 but acts via the mitochondrial Rho GTPase MIRO-1 and cytosolic $Ca^{2+}$. The fragmented mitochondria and accelerated wound closure of *fzo-1* mutants are dependent on MIRO-1 function. Genetic and transcriptomic analyzes show that enhanced mitochondrial fragmentation accelerates wound closure via the upregulation of mtROS and Cytochrome P450. Our results reveal how mitochondrial dynamics respond to cellular and tissue injury and promote tissue repair.

[1] Center for Stem Cell and Regenerative Medicine and Department of Cardiology of The Second Affiliated Hospital, Zhejiang University School of Medicine, 310058 Hangzhou, China. [2] The Zhejiang University-University of Edinburgh Institute, 718 East Haizhou Rd., Haining, 314400 Zhejiang, China. [3] Department of System Biology, School of Life Science, Wuhan University, 430072 Wuhan, China. [4] Division of Biological Sciences, Section of Cell and Developmental Biology, University of California, San Diego, 9500 Gilman Drive, La Jolla, CA 92093, USA. [5] Women's Hospital of Zhejiang University, School of Medicine Hangzhou, 310058 Hangzhou, China. [6] These authors contributed equally: Hongying Fu, Hengda Zhou. ✉email: shxu@zju.edu.cn

Mitochondria are highly dynamic organelles organized in sophisticated intracellular networks linked to their physiological roles[1]. Mitochondria can rapidly and transiently change their morphology in response to environmental changes, altering their function in energy production, $Ca^{2+}$ homeostasis, redox signaling, and metabolism[2]. Disrupted mitochondrial morphology is associated with many human diseases, including heart failure, neurodegenerative disorders, and cancer[1].

Fusion and fission of mitochondrial membranes are coordinated. The balance between fusion and fission determines the shape, size, and connectedness of mitochondria, accounting for the variation in mitochondrial morphology in different cell types[3]. The core regulators of these dynamic transitions are highly conserved dynamin-like guanosine triphosphatases (GTPases)[4]. Mitochondrial fission is conducted by the dynamin-related protein 1(DRP1), which is recruited to the mitochondrial surface in response to a variety of cues[5–8]. A sophisticated fission-machinery, including DRP1-specific adapters, actin nucleating proteins[9], and myosin[10], assemble at contact sites between the endoplasmic reticulum and mitochondria. A recent finding indicates that $Ca^{2+}$ signals can regulate a rapid mitochondrial shape transition (MiST) through DRP-1 independent but MIRO-1-dependent way[11]. Fusion, in contrast, is executed by the Mitofusins 1/2 (MFN1/2) and OPA1, which drive outer and inner mitochondrial membrane fusion, respectively[4].

Fusion contributes to mitochondrial maintenance, whereas fission causes mitochondrial fragmentation, which allows removal of irreversibly damaged mitochondria by mitophagy[12]. Fragmentation of the mitochondrial network occurs in response to cellular stress and during cell death[2]. Although mitochondrial fragmentation can cause mitochondrial dysfunction in disease, mitochondrial fragmentation also plays a positive role. For example, mitochondrial fragmentation promotes continued clearance of apoptotic cells by macrophages[13], accelerates cell proliferation[14], and regulates systemic glucose homeostasis[15]. However, the mechanisms by which fragmented mitochondria protect against cellular stresses is not clear.

Tissue injury induces coordinated responses that allow for efficient wound repair, which is important for animal survival and reproduction. For example, in different species including mammals, wounding activates intracellular, and intercellular transcriptional growth factor and chemokine cascades that regulate gene expression during tissue repair[16,17]. Cellular and tissue injury also triggers multiple damage signals, such as $Ca^{2+}$, reactive oxygen species (ROS), and ATP, which control transcription-independent wound responses to restore cellular architecture and function[18–25]. Studies in a variety of organisms suggest mitochondrial ROS (mtROS) signals can promote tissue repair[26–30]. However, the physiological roles of mitochondria in tissue damage responses and repair have not been extensively characterized.

Here we report that wounding C. elegans epithelial cells triggers rapid and reversible mitochondrial fragmentation, a process we refer to as wounding-induced mitochondrial fragmentation (WIMF). We show that enhanced mitochondrial fragmentation accelerates wound closure in vivo. WIMF is independent of the canonical DRP-1 mediated mitochondrial fission pathway but is dependent on wound-induced $Ca^{2+}$ influx and the mitochondrial Rho GTPase MIRO-1. We define a protective mechanism initiated from mitochondrial fragmentation, which functions through the upregulation of mtROS and cytochrome P450 to promote wound closure. Our studies reveal a critical role for mitochondrial morphology in response to and promoting tissue repair.

## Results

**Tissue wounding induces rapid and reversible mitochondrial fragmentation.** We visualized mitochondrial responses to acute skin wounding in C. elegans (Fig. 1a). In the lateral epidermis of late L4 or young adult animals, mitochondria are threadlike, forming elaborate branched networks that are stable over periods of tens of seconds (Fig. 1b, Supplementary Fig. 1a; Supplementary Movie 1). We observed rapid alterations in the morphology of the epidermal mitochondrial network after wounding (Fig. 1b–e). Laser wounding destroyed the local mitochondrial network within seconds (Fig. 1b, c); over the next 5–10 min, the surrounding mitochondria changed tubular shape to fragmented within 50–70 μm of the wound site (Supplementary Movies 1 and 2).

Epidermal mitochondrial fragmentation was also observed after needle wounding, both in the wounded syncytial epidermis (hyp7) and adjacent seam cells (Fig. 1d, Supplementary Fig. 1b). Mitochondria remained fragmented for several hours and returned to normal morphology and network by 24 hours post-wounding (Fig. 1d, e). Either needle or femtosecond laser triggered rapid fragmentation within $50 \pm 10$ μm of the wound site, while Micropoint UV wounding increased the fragmented region than a needle or femtosecond laser wounding (Supplementary Fig. 1c, d; Supplementary Movies 1 and 2). Together, these data show that wounded C. elegans epidermis displays a rapid and reversible change in mitochondrial morphology that we term wounding induced mitochondrial fragmentation (WIMF).

To investigate whether WIMF occurs in other tissue and cellular wound responses, we wounded the tail fin in zebrafish larvae and found widespread mitochondrial fragmentation around the wound edge 5 min after injury (Fig. 1f). We also observed similar mitochondrial fragmentation 5 min after scratch wounding of a monolayer of U2OS cells at the wounding edge (Supplementary Fig. 1e, f), suggesting WIMF is a general subcellular response to tissue wounding.

**Chronic and acute induction of mitochondrial fragmentation accelerates epidermal wound closure.** To investigate the function of mitochondrial fragmentation in epidermal wound repair, we examined actin-mediated wound closure[23]. The ring of actin polymerization at the wound site is surrounded by fragmented mitochondria (Supplementary Fig. 2a, Supplementary Movie 3). In C. elegans, mitochondrial fusion requires the outer membrane protein FZO-1 (orthologous to human MFN1/2) and the inner membrane protein EAT-3 (orthologous to human Opa1)[31]; fission requires the cytosolic protein DRP-1 (orthologous to human Drp1)[5]. fzo-1 and eat-3 null mutants are defective in mitochondrial fusion and display chronic mitochondrial fragmentation (Fig. 2a). Surprisingly, these animals displayed faster wound closure compared to the wild type (WT) (Fig. 2a, b, Supplementary Fig. 2b, c, Supplementary Movie 4). Conversely, loss of function in drp-1, which causes chronic mitochondrial elongation, did not significantly impair wound closure (Fig. 2a, b, Supplementary Fig. 2c). Depleting the mitochondrial fusion gene not only induces mitochondrial fragmentation but also results in defective ETC activity[32,33]. We observed that the oxygen consumption rate (OCR) was significantly reduced in fzo-1 and eat-3 as well as in drp-1 mutants (Supplementary Fig. 2f), suggesting the enhanced wound closure is not due to the reduced ETC activity. All these mitochondrial mutants showed normal survival post-wounding (Supplementary Fig. 2d, e). Expression of fzo-1 genomic DNA under the control of its own promoter or heat-shock promoter rescued fzo-1 mutant mitochondrial morphology and restored wound closure rates to normal (Fig. 2c, d).

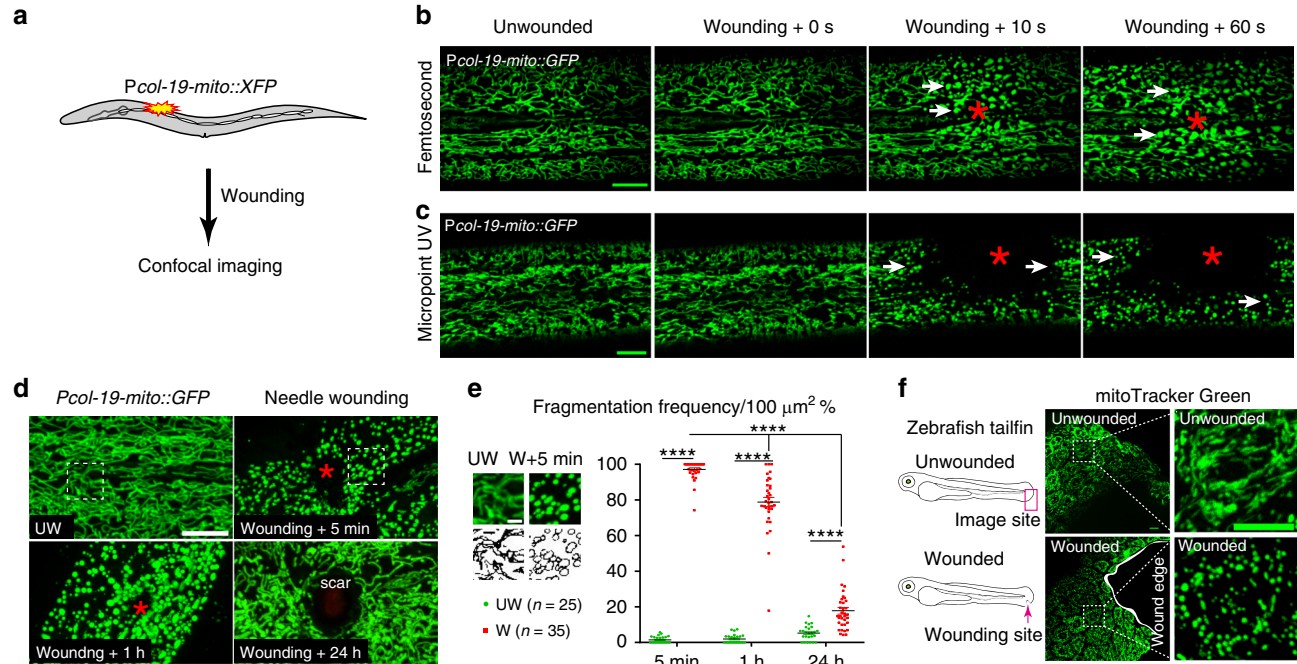

**Fig. 1 Wounding triggers rapid and reversible mitochondrial fragmentation. a** Experimental design to investigate the mitochondrial response to epidermal wounding (laser or physical damage) in *C. elegans*. Mitochondria were labeled with matrix targeting sequence from cox 8 (mito) fused with XFP (including P*col-19-mito::GFP(juEx4796)*, P*hpy-7*(y37a1b.5)-*mito::GFP(yqIs157)*[70], P*col-19-mito::mKate2(zjuSi47)*, or P*col-19-mito::dendra2(juSi271)*). **b**, **c** Laser wounding induces mitochondrial fragmentation in the epidermis. Representative confocal images of the epidermal mitochondrial network before and seconds after wounding by femtosecond laser (**b**) (see also Supplementary Movie 1, N = 5 independent experiments) and Micropoint UV laser (**c**) (see also Supplementary Movie 2, N = 7 independent experiments). P*col-19-mito::GFP(juEx4796)* was used to label mitochondria. We define mitochondrial fragmentation as a change from the interconnected tubular structure network to a rounded shape. **d** Mechanical needle wounding causes fragmentation of epidermal mitochondria, which return to normal morphology 24 hours after wounding except for a scar region at the center of the wound site. Representative confocal images of epidermal mitochondria before and after needle wounding. N = 3 independent experiments. P*col-19-mito::GFP(juEx4796)* was used to label mitochondria. Red asterisks in **b**–**d** indicate the wound site. White dashed squares indicate the zoom-in images for panel **e**. Scale bars **b**–**d**, 10 μm. **e** Quantitation of mitochondrial fragmentation frequency after needle wounding, measured in 100 μm² regions of interest (white dash square in panel **d**) 10 μm adjacent to the wound site. Top panel shows enlarged images of mitochondria in unwounded (UW, n = 25) and wounded (W, n = 35) epidermis. Scale bars, 5 μm. Bars indicate mean ± SEM. ****P < 0.0001, Two-tailed unpaired *t*-test for wounded animals. Source data are provided as a Source Data file. **f** Wounding induce mitochondrial fragmentation in zebrafish tail fin. Left, experimental design for zebrafish tail fin wounding, 3 dpf larvae were first stained with mitoTracker Green for 2 h and then were wounded using needle. N = 3 independent experiments. Right, representative confocal image of mitochondria at the edge of zebrafish tail fin, Scale bar, 10 μm.

To further test the role of mitochondrial fragmentation in wound closure, we treated WT animals with drugs that induce mitochondrial fragmentation through the inhibition of electron transport chain (ETC)[34]. In total, 2 hours treatment of young adult animals with either rotenone or antimycin A induced mitochondrial fragmentation (Fig. 2e) and accelerated actin-based wound closure (Fig. 2g). Moreover, acute treatment with Carbonyl cyanide-4-(trifluoromethoxy) phenylhydrazone (FCCP) before wounding, which also induces mitochondrial fragmentation (Fig. 2e), significantly accelerated wound closure 1 h.p.w. (Fig. 2f, g). Thus, chronic or acute mitochondrial fragmentation can accelerate epidermal wound closure.

**Mitochondrial fragmentation accelerates epidermal wound closure cell-autonomously.** We next examined whether mitochondrial fragmentation promotes wound closure in a cell-autonomous or cell-nonautonomous way by tissue-specific knockdown of *fzo-1* using the GFP nanobody mediated protein degradation (G-DEG) system (Fig. 2h)[35]. *fzo-1::GFP* knock-in animals showed interconnected tubular mitochondrial morphology and normal wound closure (Fig. 2i). Expression of G-DEG either ubiquitously or specifically in the adult epidermis reduced the FZO-1::GFP signal and caused mitochondrial fragmentation (Fig. 2i). Moreover, these animals displayed accelerated wound

closure 1 hour after injury (Fig. 2j). In contrast, the expression of G-DEG in muscles did not affect epidermal wound closure (Fig. 2j). Epidermal specific expression of *fzo-1* rescued fragmented mitochondria and suppressed faster wound closure in *fzo-1(tm1133)* animal (Fig. 2c,d). However, muscle expression of *fzo-1* could not suppress the enhanced wound closure in *fzo-1(tm1133)* animal (Fig. 2d). Collectively, these results show that mitochondrial fragmentation acts cell-autonomously in wound closure.

**WIMF is independent of canonical mitochondrial fission regulators.** To define how mitochondrial fragmentation enhances wound closure, we first examined how wounding triggers mitochondrial fragmentation. DRP-1 is essential for most mitochondrial fission[5]. Animals lacking *drp-1* displayed constitutively fused epidermal mitochondria that nevertheless fragmented after wounding (Supplementary Fig. 3a, Supplementary Movie 5). To compare the effects of WIMF, we quantified the farthest extent of mitochondrial fragmentation from the wound site (Supplementary Fig. 3b). Loss of function of *drp-1* increased the extent of WIMF compared to the WT (Supplementary Fig. 3b, c, Supplementary Movie 5), suggesting WIMF may be negatively regulated by DRP-1. Conversely, in *fzo-1* and *eat-3* mutants mitochondria are constitutively fragmented and swollen[36,37] (Supplementary Fig. 3d), and did not appreciably change in morphology after

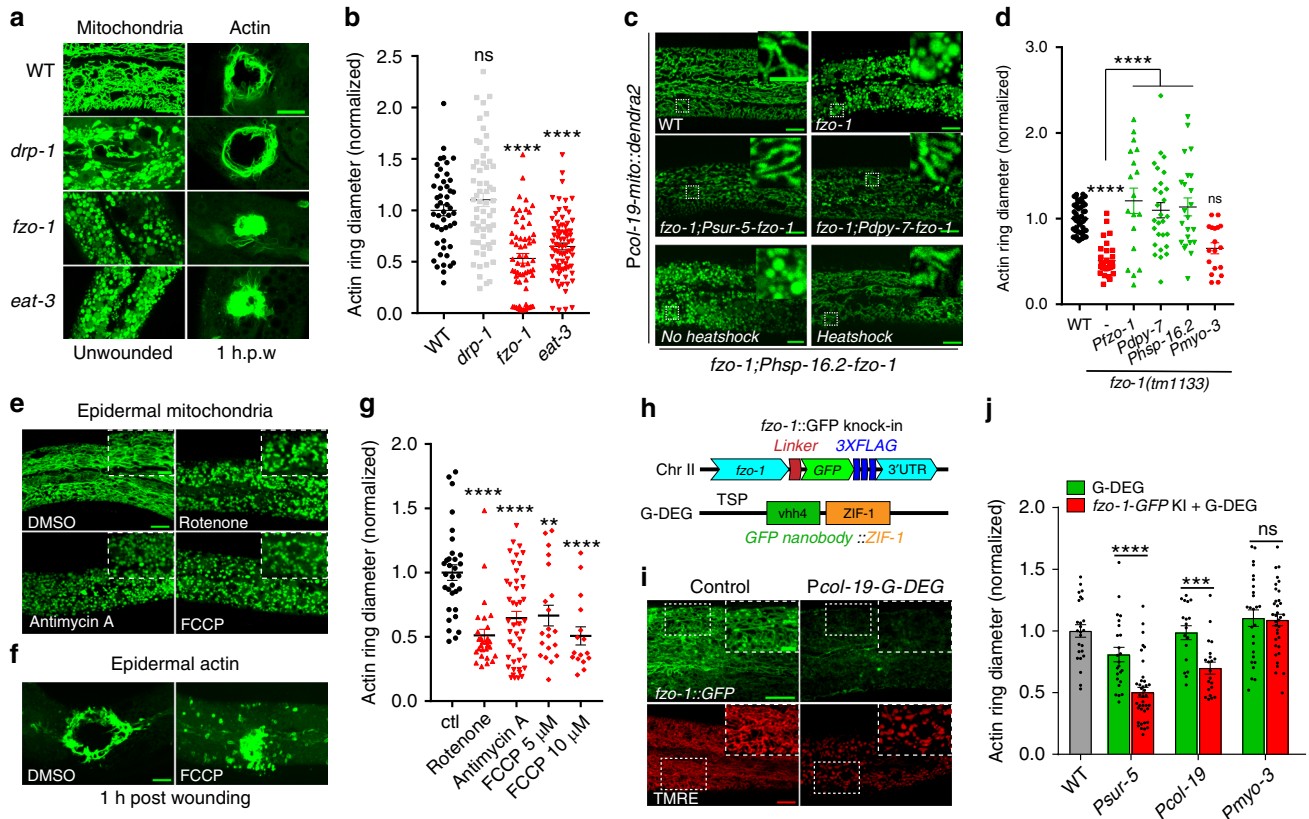

**Fig. 2 Enhanced mitochondrial fragmentation accelerates wound closure. a** Left, representative confocal images of mitochondria in WT and mutants; right, representative images of F-actin ring assembly (P*col-19-GFP::moesin(juIs352)*) at 1 h.p.w. (hour post-wounding). $N = 4$ independent experiments. **b** Quantitation of actin ring diameter in wounded animals (WT, $n = 49$; *drp-1*, $n = 61$; *fzo-1*, $n = 62$; *eat-3*, $n = 79$ animals). Bars indicate mean ± SEM, normalized to WT. ****$P < 0.0001$ (versus WT), One-way ANOVA, Dunnett's post test. **c** Representative images of mitochondria in *fzo-1(tm1133)* or tissue-specific rescued strains. **d** Quantitation of actin ring diameter (WT, $n = 51$; *fzo-1*, $n = 35$; *fzo-1;Pfzo-1-fzo-1*, $n = 18$; *fzo-1;Pdpy-7-fzo-1*, $n = 28$; *fzo-1;Phsp-16.2-fzo-1*, $n = 23$; *fzo-1;Pmyo-3-fzo-1*, $n = 23$ animals) at 1 h.p.w. Bars indicate mean ± SEM. ****$P < 0.0001$ (versus *fzo-1*), One-way ANOVA Dunnett's test. Two-tailed unpaired *t*-test for *fzo-1* and WT animals. **e** Left top, representative images of mitochondria morphology. Rotenone (100 μM), antimycin A (100 μM). and FCCP (10 μM). **f** Actin ring formation after wounding. $N = 3$ independent experiments (**e**, **f**). **g** Quantitation of actin ring diameter in WT and drug treated animals (Ctl, $n = 31$; Rotenone, $n = 28$; antimycin A, $n = 33$; FCCP 5 μM, $n = 20$; FCCP 10 μM, $n = 16$ animals) at 1 h.p.w. Bars indicate mean ± SEM. **$P = 0.016$, ****$P < 0.0001$ (versus *ctl*), One-way ANOVA Dunnett's test. **h** A scheme of *fzo-1::GFP(zju136)* and GFP nanobody mediated degradation system (G-DEG). **i** Representative images of *fzo-1::GFP* expression in animals with or without G-DEG. $N = 3$ independent experiments. **j** Quantitation of actin ring diameter after tissue-specific knockdown of *fzo-1* (WT, $n = 25$; *Psur-5;G-DEG*, $n = 39$; *Psur-5;G-DEG;fzo-1*, $n = 24$; *Pcol-19;G-DEG*, $n = 21$; *Pcol-19;G-DEG;fzo-1*, $n = 18$; *Pmyo-3;G-DEG*, $n = 26$; *Pmyo-3;G-DEG;fzo-1*, $n = 35$ animals). Bars indicate mean ± SEM. ns, $P = 0.8508$, ***$P = 0.0004$, ****$P < 0.0001$, Two-tailed unpaired *t*-test. Scale, 10 μm (**a**, **c**, **e**, **f**, and **i**) and 5 μm (zoom-in **c**, **e**, and **i**). Source data are provided as a Source Data file.

needle or laser wounding (Supplementary Fig. 3d). Furthermore, the expression and overall localization of DRP-1 were not changed in a few mins after wounding (Supplementary Fig. 3e).

We tested whether other regulators of mitochondrial morphology were required for WIMF. Mitochondrial dynamics are regulated by DRP-1 adapters, actin nucleating factors and also apoptotic components[9,10,31,38]. However, WIMF still occurred in these loss-of-function mutants (Supplementary Fig. 3f–h, details in the supplementary information). Actin polymerization and myosin affect mitochondrial fission in other systems[9,10]; however, drug treatments inhibiting either actin polymerization or non-muscle myosin, and loss of function of actin regulators did not block WIMF (Supplementary Fig. 4a–e). Together, these results suggest that WIMF does not require DRP-1 or other known regulators of mitochondrial morphology.

**WIMF spreading is dependent on mitochondrial Rho GTPase MIRO-1.** We then sought regulators of WIMF by a targeted candidate screen. We hypothesized that fragmentation signals might be sensed by the outer mitochondrial membrane (OMM)

proteins (Fig. 3a). We screened ~170 predicted OMM proteins by RNAi, most of which displayed normal epidermal mitochondrial morphology and became fragmented after wounding (Supplementary Table 4). However, RNAi knockdown of the mitochondrial Rho GTPase *miro-1* caused epidermal mitochondria to become straight and aligned, losing their typical interconnected network structure (Fig. 3b, Supplementary Fig. 5a). Moreover, in *miro-1* null mutants, WIMF is tightly restricted to the injury site (Fig. 3b, d), indicating *miro-1* is involved in the spread of WIMF. To confirm this, we performed laser wounding on mitochondria labeled with *mito::dendra2*, which can be locally converted to red fluorescence by photoconversion, making it possible to examine the morphology of single mitochondria (Fig. 3c, Supplementary Fig. 5a, Supplementary Movie 6). The distance of WIMF was reduced in *miro-1(tm1966)* mutants after wounding (Fig. 3c, d).

MIRO-1 is an outer mitochondrial membrane protein that functions as an adapter for microtubule-mediated transport[39]. GFP::MIRO-1 fusion protein was localized external to the mitochondrial matrix (Fig. 3e, Supplementary Fig. 5b) and partially co-localized with the outer mitochondrial membrane

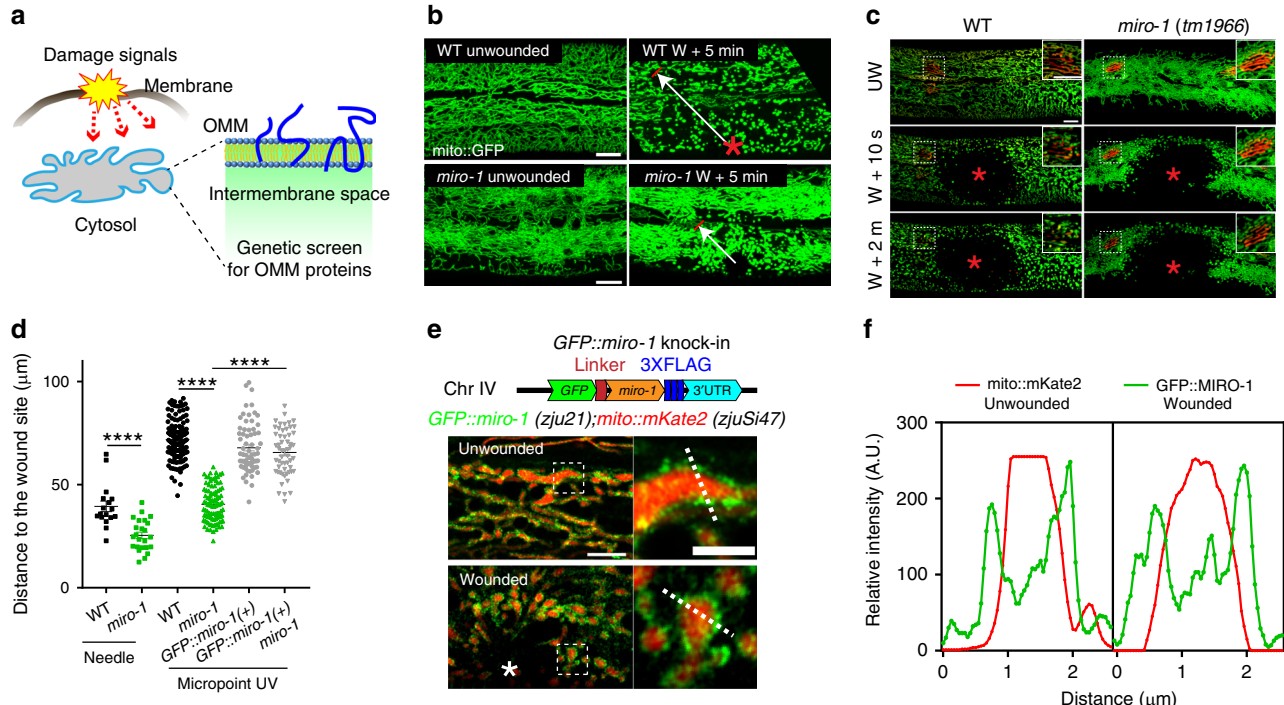

**Fig. 3 WIMF is dependent on mitochondrial Rho GTPase MIRO-1. a** Diagram of a genetic screen for outer mitochondrial membrane (OMM) proteins involved in WIMF. The candidate genes and mitochondrial morphological phenotype before and after wounding are listed in Supplementary Table 4. **b** Representative confocal images of epidermal mitochondria before and after needle wounding. $N = 5$ independent experiments. Arrows indicate the farthest extent of fragmented mitochondria from the wound site, which we used for the quantitation of distance of mitochondrial fragmentation. **c** Representative confocal images of epidermal mitochondria before and after Micropoint UV laser wounding (see also Supplementary Movie 6). Mitochondria were labeled by P*col-19-mito::dendra2(juSi271)* (**b**, **c**). Red mitochondria were photo-converted with a 405-nm laser. $N = 3$ independent experiments. Note WIMF is restricted to the injury site and does not spread to adjacent mitochondria in *miro-1(tm1966)* mutants. Scale bars (**b**, **c**), 10 μm. **d** Quantitation of the distance of fragmented mitochondria to the wound site after needle or Micropoint UV wounding in *miro-1(tm1966)* and rescued animals (WT-needle, $n = 17$; *miro-1*-needle, $n = 24$; WT-laser, $n = 144$; *miro-1*-laser, $n = 99$; GFP::*miro-1*( + )-laser, $n = 68$; *miro-1*; GFP::*miro-1*( + )-laser, $n = 56$ animals). Typically, we generated 20 arrows, as shown in panel **b** and averaged for each animal, see detail in methods. Bars indicate mean ± SEM. ****$P < 0.0001$, Two-tailed unpaired $t$-test. Source data are provided as a Source Data file. **e** Localization of GFP::MIRO-1 before and after needle wounding. Top, diagram of *GFP::miro-1* knock-in strategy. Bottom, representative confocal image of *Pcol-19-GFP::miro-1(zju21); Pcol-19-mito::mKate2(zjuSi47)* (see also Supplementary Movie 7). GFP::MIRO-1 is localized around mito::mKate2 and was remained at the wounding site immediately after wounding. Scale bar, 5 μm and 2 μm (zoom-in). $N = 2$ independent experiments. **f** Fluorescence profiles from line scan 1 and 2 illustrate the presence of GFP::MIRO-1 on a section of the mitochondrial membrane before and after needle wounding. A.U. arbitrary units. Images representative of five animals. Note, the GFP signal surrounds the mitochondrial matrix mito::mKate2 signal.

protein Tomm-20 (Supplementary Fig. 5c). After wounding, GFP::MIRO-1 localized to the wounding site and encircled the mitochondria matrix (Fig. 3e, f, Supplementary Fig. 5b, c, Supplementary Movie 7). Expression of WT *miro-1* genomic DNA specifically in the epidermis rescued the mitochondrial morphology in *miro-1(tm1966)* mutant, as did a constitutively active (CA) version of MIRO-1, but not MIRO-1 dominant-negative (DN) or EF-hand mutation (Supplementary Fig. 5d)[40]. Moreover, WIMF was also restored in *miro-1* mutant with the expression of *miro-1* genomic DNA in the epidermis (Fig. 3d). Thus, the outer mitochondrial membrane protein MIRO-1 is required for WIMF.

**Wounding-induced microtubule depolymerization is not required for WIMF.** MIRO-1 is a crucial regulator of mitochondrial motility and distribution along the microtubule in many cell types[39,41]. Microtubule stabilization also plays critical roles in wound closure[42]. We thus examined whether WIMF might involve altered microtubule dynamics after wounding. In the unwounded epidermis, mitochondria and microtubules were closely associated and relatively stable over periods of several

minutes (Supplementary Fig. 4f, Supplementary Movie 8). In *miro-1(tm1966)* mutant mitochondria were more localized to the perinuclear region (Supplementary Fig. 4f); however, microtubule distribution and dynamic were normal (Supplementary Fig. 4f, Supplementary Movie 8), suggesting loss of function in *miro-1* does not affect epidermal microtubule dynamics.

We found wounding triggers not only immediate mitochondrial fragmentation but also rapid local microtubule depolymerization (Supplementary Fig. 4g, Supplementary Movie 9). Moreover, mitochondrial fragmentation happens before microtubule depolymerization (Supplementary Fig. 4g). *miro-1* mutants displayed normal microtubule depolymerization in response to wounding but more restricted mitochondrial fragmentation (Supplementary Fig. 4g Supplementary Movie 10). Treatment of young adult animals with Taxol or Colchicine, which stabilize and destabilize microtubule respectively, did not affect either mitochondrial morphology or WIMF (Supplementary Fig. 4h). Furthermore, knock out of microtubule dynamics regulators Kinesin/KLP-7 and PTRN-1 had no apparent effects on WIMF (supplementary Fig. 4i). Thus, microtubule stability or dynamics is not necessary for MIRO-1 regulated WIMF.

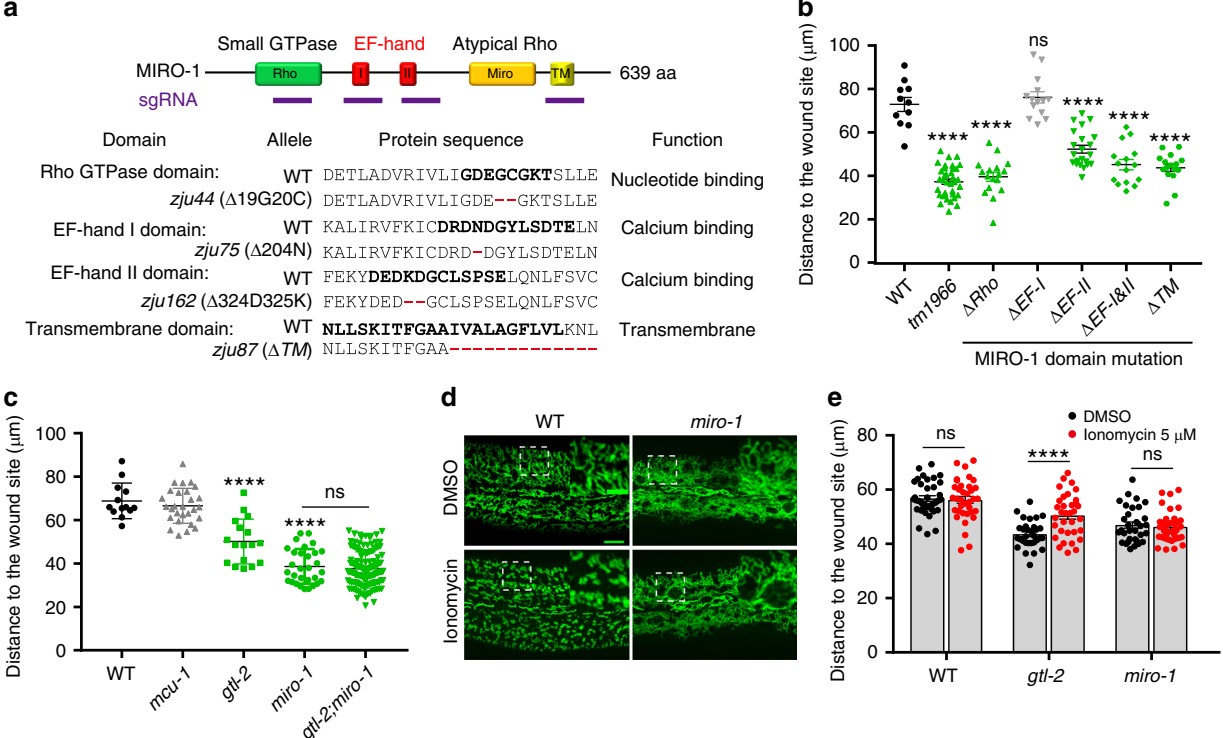

**Fig. 4 WIMF is dependent on wound-induced Ca²⁺-MIRO-1 signaling. a** Top, experimental design to disrupt the function of MIRO-1 protein domains using CRISPR-Cas9 mediated mutagenesis. Bottom, *miro-1* in-frame mutation alleles. Sequences in bold are key residues in each domain[40]. Details of indel mutations are in Supplementary Table 5. **b** Quantitation of the distance of the fragmented mitochondria to the injury site shown on the left in different animals (WT, n = 11; tm1966, n = 30; ΔRho, n = 19; ΔEF-I, n = 14; ΔEF-II, n = 22; ΔEF-I&II, n = 15; ΔTM, n = 16 animals). Note, mutations of the Rho, EF-hand II and TM domains show reduced spreading of WIMF compared to the WT and similar to miro-1(tm1966) animal, indicating that these mutations affect MIRO-1's function. Bars indicate mean ± SEM. ns, P = 0.8321, ****P < 0.0001 (versus WT), One-way ANOVA Dunnett's test. **c** Quantitation of the distance of the fragmented mitochondria to the injury site in TRPM/gtl-2(n2618), mcu-1(ju1154), and gtl-2(n2618);miro-1(zju19) mutant (WT, n = 13; gtl-2, n = 17; miro-1, n = 36; gtl-2;miro-1, n = 143 animals). miro-1(zju19) and miro-1(zju40) frameshift mutations were generated by CRISPR-Cas9 mediated mutagenesis in gtl-2 background. Bars indicate mean ± SEM. ns, P = 0.9558 (versus miro-1), two-tailed unpaired t-test; ns, P = 0.9558, ****P < 0.0001 (versus WT), One-way ANOVA Dunnett's test. **d** Representative confocal image of mitochondrial morphology after treatment with Ca²⁺ ionophore ionomycin in both WT and miro-1 mutants. Scale bars, 10 μm, and 5 μm (zoom-in). N = 3 independent experiments. **e** Quantitation of the distance of the fragmented mitochondria to the injury site in gtl-2(n2618), miro-1(tm1966) mutant treated with low concentration of ionomycin (2.5 μM treated for 1 h) (WT-DMSO, n = 37; WT-ionomycin, n = 36; gtl-2-DMSO, n = 31; gtl-2-ionomycin, n = 34; miro-1-DMSO, n = 32; miro-1-ionomycin, n = 35). Bars indicate mean ± SEM. ns, P = 0.7435 (WT) or 0.6382 (miro-1), ***P < 0.0001, Two-tailed unpaired t-test. Source data are provided as a Source Data file.

**Ca²⁺-MIRO-1 signaling is required for the spreading of WIMF.** MIRO-1 has a C-terminal transmembrane (TM) domain for OMM targeting and two Rho GTPase domains flanking two Ca²⁺ sensing EF-hand domains (Fig. 4a). To understand how MIRO-1 might regulate WIMF, we generated domain-specific alleles of the endogenous *miro-1* locus (Fig. 4a, Supplemental Table 5). Rho, EF-hand, and TM domain mutants displayed similar mitochondrial morphology as in *miro-1(tm1966)* before wounding (Supplementary Fig. 5e), suggesting these domains are required for MIRO-1 to maintain mitochondrial morphology. Moreover, WIMF is more restricted in the animals with mutated Rho, EF-hand, or TM domains (Fig. 4b, Supplementary Fig. 5f, Supplementary Movie 11), indicating that outer mitochondrial membrane localization of MIRO-1 and cytosolic Ca²⁺ binding to EF-hand is important for the WIMF.

Epidermal wounding also triggers rapid elevation of cytosolic Ca²⁺[23,24,43]; in the *C. elegans* skin, this is dependent on the TRPM/*gtl-2* channel[23]. Ca²⁺ uptake into mitochondria mediated by the MCU-1 channel is also necessary for actin-based wound closure[26]. Moreover, MIRO-1 has been shown to be required for a cytosolic Ca²⁺ induced mitochondrial shape transition in cultured cells[11]. We tested whether wound-induced early Ca²⁺

signals were involved in WIMF. In *gtl-2(n2618)* or *mcu-1(ju1154)* loss of function mutants, which display reduced cytosolic or mitochondrial Ca²⁺ uptake after wounding respectively, epidermal mitochondria displayed normal steady-state morphology (Supplementary Fig. 5f, Supplementary Movie 12). *gtl-2(n2618)* mutant showed much reduced mitochondrial fragmentation, whereas *mcu-1(ju1154)* mutant underwent typical mitochondrial fragmentation (Fig. 4c, Supplementary Fig. 5f, Supplementary Movie 12). *gtl-2 miro-1* double mutants resembled *miro-1* single mutants in the distance of WIMF (Fig. 4c). Together, these data suggest that the GTL-2 mediated cytosolic Ca²⁺ signal acts in the same pathway as MIRO-1 to drive WIMF.

To further test this hypothesis, we treated animals with ionomycin to elevate cytosolic Ca²⁺ (Supplementary Fig. 5g) and observed widespread mitochondrial fragmentation in WT animals but not in *miro-1* mutants (Fig. 4d), suggesting MIRO-1 is required for elevated Ca²⁺ to induce mitochondrial fragmentation, consistent with findings in cultured cells[11]. Furthermore, the elevation of Ca²⁺ by low doses of ionomycin rescued WIMF in *gtl-2* mutants but not in *miro-1* animals (Fig. 4e, Supplementary Movie 13), suggesting wounding-induced Ca²⁺ is necessary for WIMF and acts via MIRO-1.

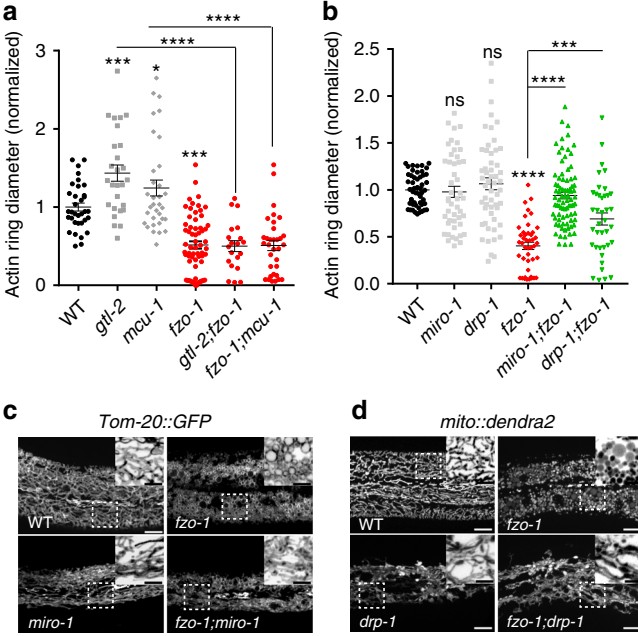

**Fig. 5 Mitochondrial fragmentation acts downstream of $Ca^{2+}$-MIRO-1 to accelerate wound closure. a** Loss of function in the fusion gene *fzo-1* suppresses wound closure defects of *gtl-2(n2618)* and *mcu-1(ju1154)* mutants. Quantitation of actin ring diameter in WT and mutants 1 h after needle wounding (WT, *n* = 50; *gtl-2*, *n* = 27; *mcu-1*, *n* = 33; *fzo-1*, *n* = 64; *gtl-2;fzo-1*, *n* = 20; *fzo-1;mcu-1*, *n* = 33 animals). *gtl-2* and *mcu-1* mutants display delayed actin ring closure, which is suppressed in double mutants with *fzo-1(tm1133)*. The WT actin ring diameter was normalized to 1 and mutants normalized to WT. Bars indicate mean ± SEM. *$P$ = 0.0328, ***$P$ = 0.0001, ****$P$ < 0.0001, One-way ANOVA Dunnett's test (single mutant versus WT). two-tailed unpaired *t*-test (versus *mcu-1* or *gtl-2*). **b** Quantitation of actin ring diameter in the mitochondrial mutants (WT, *n* = 34; *drp-1*, *n* = 54; *miro-1*, *n* = 47; *fzo-1*, *n* = 44; *miro-1;fzo-1*, *n* = 82; *fzo-1; drp-1* RNAi, *n* = 39 animals). *drp-1* partially suppresses *fzo-1* while *miro-1* significantly suppresses the *fzo-1* in enhanced wound closure. Bars indicate mean ± SEM. ns, $P$ = 0.7753 (*drp-1*) or 0.9893 (*miro-1*), ***$P$ = 0.0002, ****$P$ < 0.0001, two-tailed unpaired *t*-test (versus *fzo-1*). One-way ANOVA Dunnett's test (single mutant versus WT). Source data are provided as a Source Data file. **c** Representative confocal images of mitochondrial morphology in *fzo-1* and *miro-1;fzo-1* mutants. *Pcol-19-Tomm-20::GFP (zjuSi48)* transgenic animals label the mitochondria. *fzo-1* mutants display fragmented and round shaped mitochondria. *miro-1;fzo-1* double mutants have tubular mitochondria, similar to *miro-1* single mutants. *N* = 3 independent experiments. **d** Representative confocal images of mitochondrial morphology in *fzo-1* and *drp;fzo-1* mutants. Mitochondria were labeled with *mito::dendra2(juSi271)*, the mitochondrial morphology of *drp-1;fzo-1* mimics *drp-1*, without fragmented mitochondria. Note, *drp-1* suppress *fzo-1* in mitochondrial morphology. Scale (**c**, **d**), 10 μm and 5 μm (zoom-in image).

## Mitochondrial fragmentation acts downstream of $Ca^{2+}$ signaling in wound closure.

We next asked how mitochondrial fragmentation accelerates wound closure. Loss of function in either the $Ca^{2+}$ channel TRPM/GTL-2 or the mitochondrial $Ca^{2+}$ uniporter MCU-1 inhibits wound closure[23,26]. We thus performed epistasis analysis of *fzo-1* with $Ca^{2+}$ and the mitochondrial $Ca^{2+}$ pathway. We found that loss of function in *fzo-1* suppressed the low post-wounding survival of *gtl-2* (Supplementary Fig. 2d) and restored its wound closure to normal (Fig. 5a). *fzo-1* also rescued the defective wound closure in the *mcu-1* mutant (Fig. 5a). Thus, constitutive fragmentation of mitochondria in *fzo-1* mutants bypasses the need for cytosolic or mitochondrial $Ca^{2+}$ signals in wound closure, consistent with $Ca^{2+}$ acting upstream of WIMF (Fig. 4).

## MIRO-1 is required for the accelerated wound closure of mutants with fragmented mitochondria.

We then asked whether the enhanced wound closure in *fzo-1* is due to fragmented mitochondria. Although *miro-1* mutation did not significantly affect wound closure or post-wounding survival (Fig. 5b; Supplementary Fig. 5h), the faster wound closure of the *fzo-1* mutant was suppressed by loss of function in *miro-1* (Fig. 5b). Surprisingly, mitochondrial morphology of *miro-1; fzo-1* mutants resembled that of *miro-1* single mutants (Fig. 5c), indicating that *miro-1* is required for the fragmented mitochondrial morphology of *fzo-1* mutant and that mitochondrial fragmentation is required for *fzo-1*'s faster wound closure. To test this further, we made *drp-1 fzo-1* double mutants, which displayed a hyper-fused mitochondrial morphology resembling that of *drp-1* (Fig. 5d). *drp-1* also significantly suppressed the faster wound closure phenotype of *fzo-1* animals (Fig. 5b).

## Oxidative signaling genes are upregulated by wounding and by mitochondrial fragmentation.

To understand the genetic basis of the enhanced wound closure in animals with fragmented mitochondria, we analyzed the transcriptomes of wounded WT animals and *fzo-1* mutants, using unwounded WT animals as the control (Fig. 6a). A large number of differentially expressed genes (DEGs) were induced in wounded animals compared to unwounded ones (Supplementary Fig. 6a, Supplementary Data 1). DAVID gene ontology (GO) analysis found that most such DEGs were enriched in the mitochondria and mitochondrial biogenesis (Supplementary Fig. 6b, c). We hypothesized that faster wound closure of *fzo-1* mutants might reflect the elevated expression of genes induced by wounding. We compared RNA seq data in wounded WT animals and *fzo-1* mutants and identified ~2200 and ~800 DEGs, respectively (adjusted *p*-value < 0.05) (Fig. 6a). Strikingly, 216 genes differentially expressed after skin wounding showed similar differential expression in *fzo-1* (Fig. 6b, Supplementary Data 2). GO analysis of the 216 overlapping genes revealed enrichment for oxidation-reduction and metabolism-related terms (Fig. 6c), suggesting loss of function of *fzo-1* or wounding affects the expression of genes involved in oxidative signaling. These included cytochrome P450 (*cyp*) family genes, dehydrogenases (*dhs*), and glutathione S-transferases (*gst*) (Fig. 6d, Supplementary Fig. 6d). Quantitative PCR of selected genes revealed similar changes as found in RNA seq (Fig. 6e–l).

## Enhanced wound closure in *fzo-1* is dependent on Cytochrome P450 and mtROS.

How might oxidative signals affect wound closure in animals with fragmented mitochondria? To address this question, we examined CYP genes upregulated after wounding and in *fzo-1* mutants. RNAi knockdown of either *cyp-13A8* or *cyp-13A12* strongly inhibited actin ring formation at wound sites in both WT and *fzo-1* mutant (Fig. 6m, n). Importantly, in animals forming a visible actin ring, RNAi knockdown of *cyp-13A8*, which is highly induced after wounding and in *fzo-1* (Fig. 6d), significantly inhibits actin ring closure (Fig. 6o). We observed *cyp-13A8* mRNA expression was significantly increased a few minutes after wounding (Supplementary Fig. 6e), the time correlates with the rapid actin polymerization at the wound site (Supplementary Fig. 6e). *cyp-13A8* is expressed in many tissues, including epidermis (Supplementary Fig. 6f). Overexpression of *cyp-13A8* in epidermis either using its own promoter or *col-19* promoter accelerated wound closure (Fig. 6p), suggesting CYP-13A8 acts cell-autonomously to promote wound closure.

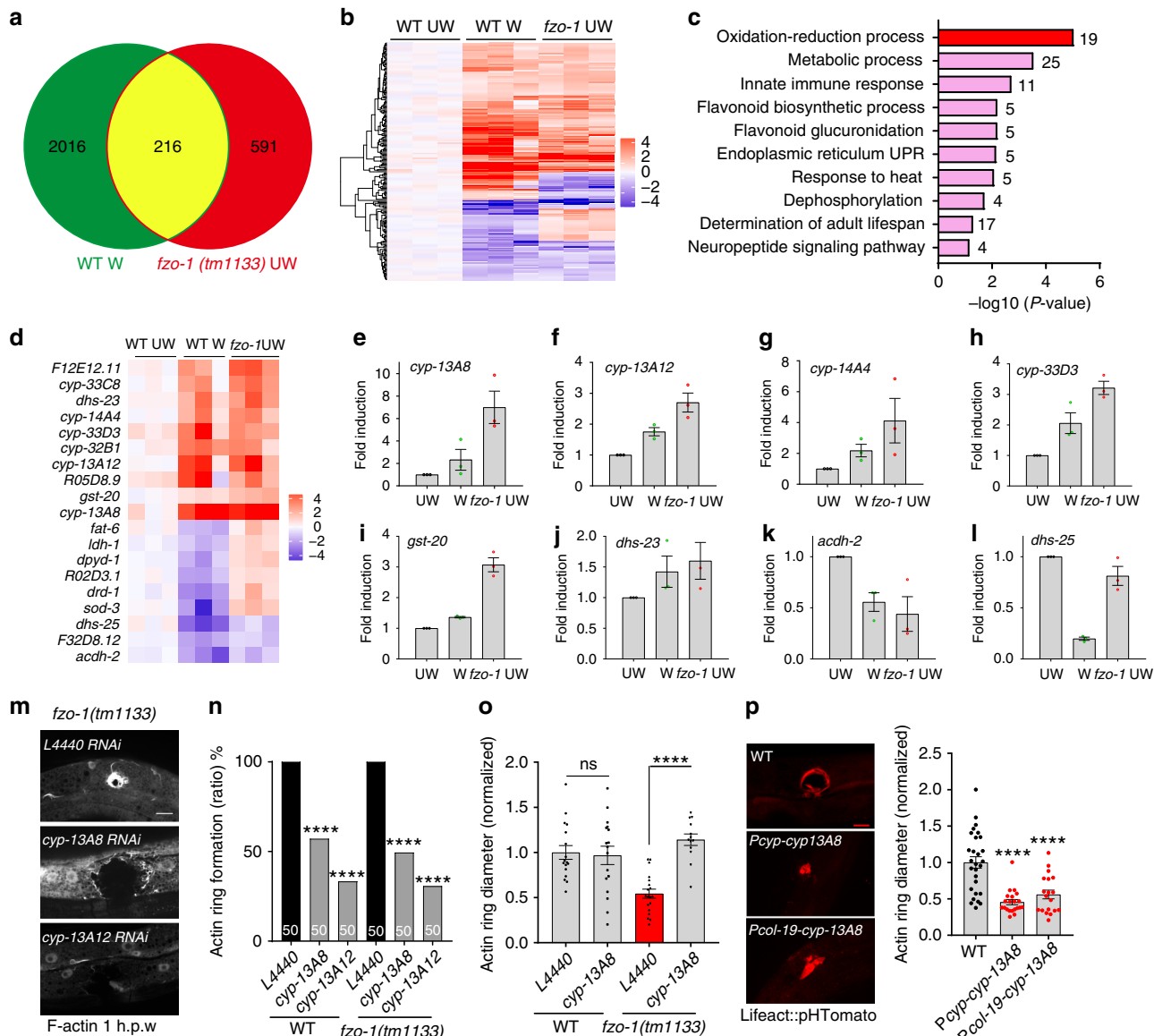

**Fig. 6 The upregulation of oxidative signals in *fzo-1* mutants is required for faster wound closure. a** Venn diagram of differentially expressed genes (DEGs) in WT needle wounded and *fzo-1(tm1133)* unwounded animals, compared to WT unwounded controls, as measured by RNA-seq (Benjamini-Hochberg adjusted *P* value <0.001). See also Supplementary Data 1. **b** Gene expression heatmap of 216 DEGs shared by WT wounded and *fzo-1* unwounded animals as described in **a**. **c** DAVID biological process GO terms of upregulated and downregulated genes in the 125/216 overlapping DEGs (Benjamini-Hochberg adjusted *P* value <0.001). See also Supplementary Data 2. **d** Heatmap of 19 overlapping oxidation-reduction processes genes. **e–l** Quantification of oxidation-reduction process-related genes in the WT, wounded, and *fzo-1* animals as determined by qPCR. In both WT wounded and *fzo-1* mutant, *cyp-13A8*, *cyp-13A12*, *cyp-14A4*, *cyp-33D3*, *gst-20*, *dhs-23*, and *dhs-28* were upregulated while *acdh-2*, *dhs-25* are down-regulated. N = 3 independent experiments; Bars indicate mean ± SEM. **m** Representative confocal images of actin ring formation 1 h.p.w. RNAi knockdown of *cyp-13A8* and *cyp-13A12* suppresses the accelerated actin ring formation in *fzo-1(tm1133)*. Scale bar, 10 μm. N = 3 independent experiments. **n** Quantitation of acti**n** ring formation percentage in the WT and *fzo-1(tm1133)* double mutants with RNAi knockdown *cyp-13A8*, and *cyp-13A12* genes. N = 50, ****P < 0.0001, versus WT or *fzo-1(tm1133)*, Two-sided Fisher's exact test. **o** Quantitation of actin ring diameter in the WT and *fzo-1(tm1133)* double mutants with RNAi knock down *cyp-13A8* (WT-L4440, n = 17; WT-*cyp-13A8*, n = 17; *fzo-1*-L4440, n = 19; *fzo-1*;*cyp-13A8*, n = 13 animals). Bars indicate mean ± SEM, ns, *P* = 0.803, ****P < 0.0001, versus WT or *fzo-1(tm1133)*. two-tailed unpaired *t*-test. **p** Overexpression of *CYP-13A8* in epidermis accelerates wound closure. Left, representative actin ring images, Scale bar, 10 μm. Right, quantitation of actin ring diameter 1 h.p.w (WT, n = 28; Pcyp-13A8-cyp-13A8, n = 21; Pcol-19-cyp-13A8, n = 19 animals). Bars indicate mean ± SEM, ****P < 0.0001, versus non transgenic animals. One-way ANOVA Dunnett's test. Source data are provided as a Source Data file.

CYPs can generate reactive oxygen species (ROS)[44,45], and CYP expression is subject to feedback regulation by mtROS[46]. Therefore, we examined mitochondrial ROS (mtROS) production in the fragmented mitochondrial mutants by staining with either the ROS sensor mitoSox or the genetically encoded sensor mito::cpYFP[47]. Both *fzo-1(tm1133)* and *eat-3(tm1107)* mutants

displayed significantly increased mitoSox fluorescence (Fig. 7a) and mito::cpYFP intensity (Supplementary Fig. 7a), whereas *drp-1 (tm1108)* mutants showed reduced mitoSox staining (Fig. 7a). Treatment of animals with mitochondrial fragmentation inducers Rotenone or FCCP significantly enhanced the mitoSox signal (Supplementary Fig. 7b) and also induced CYP expression

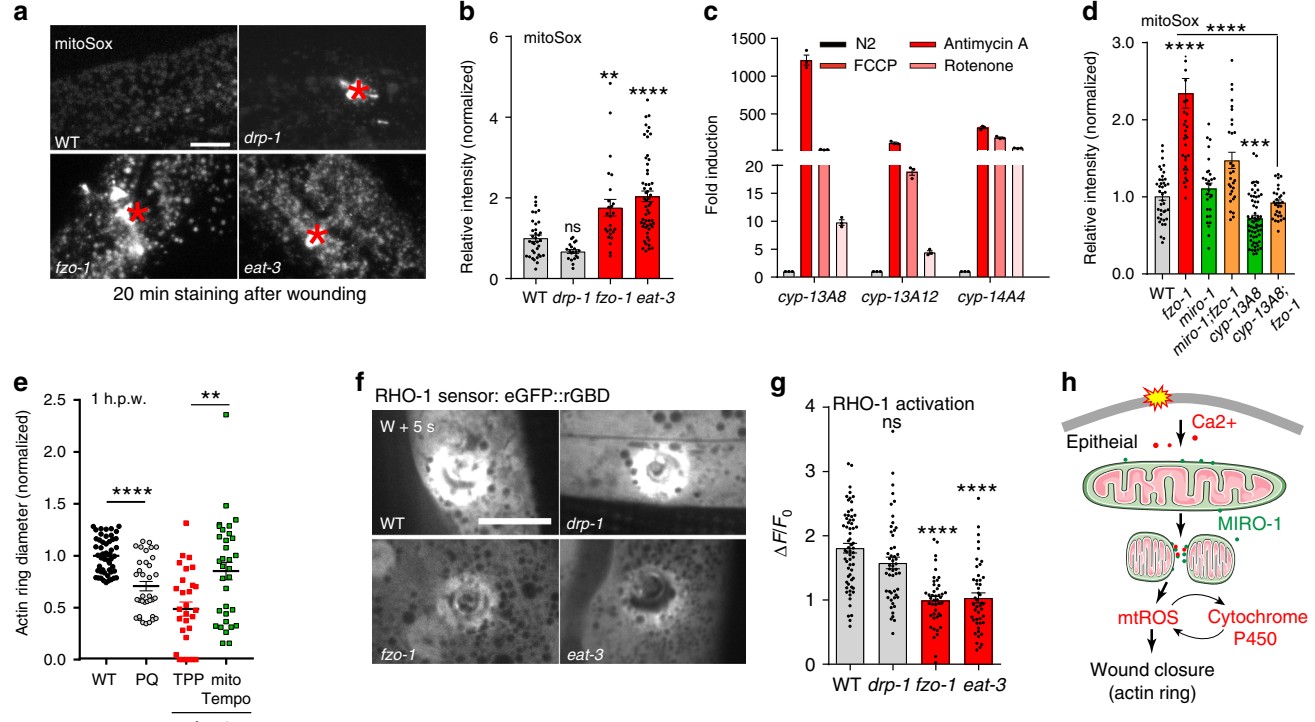

**Fig. 7 Accelerated wound closure of *fzo-1* mutant is dependent on mtROS. a** Representative confocal images of mitoSox staining in WT and mutants. **b** Representative confocal images of mitoSOX staining in WT, *drp-1(tm1108)*, *fzo-1(tm1133)*, and *eat-3(tm1107)* mutants 20 min after needle wounding. Red asterisks indicated wound sites. Right, Quantitation of mitoSOX intensity, normalized to WT (WT, $n = 35$; *drp-1*, $n = 20$; *fzo-1*, $n = 25$; *eat-3*, $n = 56$ animals). Bars indicate mean ± SEM, ns, $P = 0.3413$, **$P = 0.0016$, ****$P < 0.0001$, One-way ANOVA Dunnett's test (versus WT). **c** qPCR analysis of Cytochrome P450 genes after treatment antimycin A, Rotenone and FCCP. $N = 3$ independent experiments. Bars indicate mean ± SEM. **d** Quantitation of mitoSOX intensity, normalized to WT (WT, $n = 40$; *fzo-1*, $n = 38$; *miro-1*, $n = 29$; *fzo-1;miro-1*, $n = 30$; *fzo-1;cyp-13A8*, $n = 30$; *cyp-13A8*, $n = 62$ animals). Bars indicate mean ± SEM, ns, $P = 0.2472$, ***$P = 0.0005$, ****$P < 0.0001$, versus WT or *fzo-1* animal. two-tailed unpaired *t*-test. **e** Quantitation of F-actin ring diameter in TPP and mitoTempo treated worms (WT, $n = 51$; PQ, $n = 35$; *fzo-1*-TPP, $n = 29$; *fzo-1*-mitoTempo, $n = 33$ animals). Paraquat (PQ) treatment as a control. Bars indicate mean ± SEM, **$P = 0.0015$, ****$P < 0.0001$, two-tailed unpaired *t*-test. **f, g** Mitochondrial fusion mutants reduced local activation of RHO-1 small GTPase at wound sites (see also Supplementary Movie 14). **f** Representative confocal images of RHO-1 GTPase sensor eGFP::rGBD in the adult in WT or *drp-1, fzo-1,* and *eat-3* mutant. *Pcol-19-eGFP::rGBD(juEx3025)* strain were used for wounding and imaging. Scale bars (**a**, **f**), 10 μm. **g** Quantitation of eGFP::rGBD fluorescent intensity change in WT and mutants after wounding (WT, $n = 52$; *drp-1*, $n = 51$; *fzo-1*, $n = 44$; *eat-3*, $n = 46$ animals). Bars indicate mean ± SEM, ns, $P = 0.1448$, ****$P < 0.001$, One-way ANOVA Dunnett's test. Source data are provided as a Source Data file. **h** A model for how mitochondrial fragmentation promotes epidermal wound closure.

(Fig. 7c). Conversely, overexpression of *cyp-13A8* in the epidermis enhanced mitoSox fluorescence while *cyp-13A8(zju212)* knock out mutants (Supplementary Fig. 7b) displayed lower mitoSox signal (Supplementary Fig. 7c), suggesting CYPs are required and sufficient to increase mtROS. Together, these results suggest that mtROS and CYP-13A8 display positive feedback regulation that maintains elevated mtROS levels after wounding.

We next determined whether the elevated mtROS in *fzo-1* mutants contributes to the enhanced wound closure. Loss of function in *miro-1* suppressed the elevated mitoSox signal in the *fzo-1* mutant (Fig. 7d), consistent with suppression of mitochondrial fragmentation and enhanced wound closure in *fzo-1* (Fig. 5b). *cyp-13A8 fzo-1* double mutants also showed significantly reduced mitoSox signal compared to *fzo-1* single mutants (Fig. 7d), suggesting *cyp-13A8* is required for the elevated mtROS levels in *fzo-1* mutants. Treatment of *fzo-1* or *eat-3* mutants with antioxidants NAC or mitoTempo not only reduced mtROS levels (Supplementary Fig. 7d) but also suppressed the faster wound closure (Supplementary Fig. 7e, f, Fig. 7e), whereas mitochondrial specific prooxidant paraquat enhanced wound closure (Fig. 7e). mitoTempo treatment reduced the post-wound survival (Supplementary Fig. 7g) and delayed wound closure (Supplementary Fig. 7h), and these effects could be reverted by the addition of

Rotenone (Supplementary Fig. 7h), suggesting elevated mtROS is required and sufficient to accelerate wound closure. We then tested whether mitochondrial fragmentation affects local activation of RHO-1, which is an inhibitory GTPase that regulating actin filaments formation in wound closure. Both fragmented mitochondrial mutants *fzo-1* and *eat-3* displayed reduced RHO-1 genetic sensor eGFP::rGBD intensity after wounding (Supplementary Fig. 7i, j, Supplementary Movie 14), suggesting mitochondrial fragmentation mutants inhibit the local action of RHO-1 in vivo. Together, these results suggest that enhanced wound closure in fragmented mitochondrial mutants is due to elevated mtROS signal and inhibition of RHO-1 activity.

## Discussion

We have shown that epidermal injury triggers rapid and reversible fragmentation of mitochondrial morphology, wounding-induced mitochondrial fragmentation (WIMF), that promotes tissue repair through oxidative-reduction and mtROS signaling. Notably, WIMF is independent of the canonical fission regulator DRP-1, but requires the $Ca^{2+}$-sensitive mitochondrial Rho GTPase MIRO-1. We find that mitochondrial fragmentation triggers mtROS production and induces expression of oxidative

signaling genes such as CYP-13A8, which acts together to maintain an elevated protective level of mtROS. Thus, our findings demonstrate that mitochondrial fragmentation is beneficial in promoting tissue repair in vivo.

Mitochondria form a highly dynamic tubular network within cells such as the *C. elegans* epidermis, reflecting a balance of fusion and fission events that are linked to the energetic and oxidative metabolic requirements of the cell. The master regulator of fission is DRP1, which constricts tubular mitochondria to facilitate division in response to cellular signals or mechanical force[48]. In contrast, WIMF does not depend on DRP-1 but instead requires the mitochondrial Rho GTPase MIRO-1. Interestingly, Nemani et al. recently reported that elevated cytosolic $Ca^{2+}$ induces a mitochondrial shape transition (MiST) in HeLa and MEF cells, dependent on MIRO1 but not DRP1[11]. While WIMF and MiST may share some mechanistic overlap, we note that the $Ca^{2+}$ induced MiST happens over several minutes[11], whereas WIMF happens in seconds (see detail comparison in Supplementary Table 6). Possibly the amplitude or local nature of the wounding-induced $Ca^{2+}$ transient is sufficient to trigger a more rapid fragmentation response. In both situations, $Ca^{2+}$ binding of MIRO-1 is critical, although the mechanism by which $Ca^{2+}$-activated MIRO1 can cause mitochondrial fragmentation remains to be determined.

Tissue damage is signaled at a cellular level via damage-associated molecular patterns (DAMPs)[49] or wound response signals such as $Ca^{2+}$ or ATP[50]. In Drosophila epidermis and zebrafish tail fin, $Ca^{2+}$ stimulates DUOX2 dependent membrane production of ROS that recruit immune cells to the injury site[24,25,43]. However, $Ca^{2+}$ mediates diverse biological processes unrelated to tissue injury, raising the question of how organisms distinguish between wound-induced and physiological $Ca^{2+}$ transients. Previous studies have shown that wounding triggers mtROS production, mediated by MCU-dependent mitochondrial $Ca^{2+}$ uptake, and that mtROS can promote wound healing[26]. Here we find that MIRO-1 is another key downstream target of $Ca^{2+}$, independent of MCU-1 dependent $Ca^{2+}$ uptake into mitochondria. Thus, wounding-induced $Ca^{2+}$ signals may act via multiple effectors to generate a protective mtROS cascade in regulating tissue repair.

We have shown that WIMF occurs in a variety of cellular contexts, from in vitro scratch assays to zebrafish tail fin wounding. It will be valuable in the future to dissect whether similar pathways involving MIRO1, CYPs, and oxidative signals function in these other wound repair paradigms. Intriguingly, MIRO1 is required for MEF cell migration in vitro, at both collective and single-cell level[51], potentially relevant to our finding that *miro-1* is required for accelerated wound closure in the *fzo-1* mutant, suggesting $Ca^{2+}$-MIRO-1 signals may play a conserved role in responding to and regulating mitochondrial activity in tissue repair.

We find that wounding and mitochondrial fragmentation induce expression of cytochrome P450 monooxygenases, a large family of enzymes that generate mtROS[44,46]. Moreover, overexpression of *cyp-13A8* in epidermis enhances mtROS and promote wound closure. CYPs respond to wounding and accelerate wound epithelization in diabetic mice[52], hairless mouse ear[53], and even in plants[54]. In this study, we demonstrated that CYPs are upregulated in the fragmented mitochondrial animals and are required for actin-based wound closure, suggesting CYPs may be vital in mediating oxidative signals that promote damage repair. By uncovering a link between mitochondrial fragmentation, oxidative signaling and tissue repair, our work may open a door to novel therapeutic interventions based on mitochondrial signals. For example, hyperbaric oxygen therapy has long been used to accelerate wound repair in chronic diabetic wounds and is

thought to act in part by increasing oxidative metabolic stress[55,56]. Our results raise the possibility that hyperbaric oxygen treatment activates mitochondrial oxidative metabolic cascades, culminating in high levels of mtROS that protect tissues from damage.

## Methods

**C. elegans strains and genetics**. All *C. elegans* strains were maintained by standard methods at 20–22.5 °C on nematode growth medium (NGM) agar plates seeded with *E. coli* OP50. New strains were constructed using standard procedures, and genotypes confirmed by PCR and sequencing. All the strains used are listed in Supplementary Table 1.

**Constructs and transgenic worms**. Epidermal mitochondria were labeled with the mitochondrial target sequence of Cytochrome C oxidase VIII fused to GFP or dendra2 or mKate2 under the control of the *col-19* promoter. Extrachromosomal array transgenic worms were made by injection of constructs at 10 ng/μl with 50 ng/μl co-injection marker (Pttx-3-RFP). Single-copy insertion of Pcol-19-mito:: dendra2(juSi271) was made by Mos-SCI[57]. The single-copy insertions Pcol-19-lifeact::PHtomato(zjuSi22) I, Pcol-19-mito::mKate2(zjuSi47) II, Pcol-19-Tomm-20:: GFP(zjuSi48) I were made by CRISPR-Cas9 based insertion method[58]. New transgenic strains are listed in Supplementary Table 1. All constructs used in this study are listed in Supplementary Table 2.

**Outer mitochondrial membrane proteins screen**. RNAi was carried out as follows. Briefly, WT animals carrying the mitochondrial marker Pcol-19-mito::dendra2 (juSi271) were grown to the L4 stage and then fed RNAi bacteria for 12 h. Young adult animals were imaged on the spinning disk confocal microscope before and after needle wounding (Andor 100×, NA 1.46 objective). We screened *C. elegans* orthologs of about 170 putative OMM proteins in the mitoCarta database (https://www.broadinstitute.org/files/shared/metabolism/mitocarta/human.mitocarta2.0.html)[59]. Candidate genes are listed in Supplementary Table 4.

**GFP fusion protein knock-in**. GFP::miro-1(zju21), mKate2::miro-1(zju211), fzo-1:: GFP(zju136) knock-in mutations were generated using CRISPR-Cas9 method[60]. Briefly, repair templates were generated into pDD282 plasmid by Gibson assembly. We injected sgRNAs, repair template, and dpy-10 sgRNA (as selection marker) into N2 animals. Roller animals were heat-shocked to remove markers. All primers and sgRNAs are listed in Supplementary Table 3.

**Drug treatment**. All drugs were added to the bacterial lawn from a high concentration stock and allowed to dry for 1–2 hr at room temperature before transferring the young adult animals. Rotenone (Sigma R8875) was dissolved in DMSO to make 40 mM stock solutions; mitoTempo (Sigma, SML0737), triphenylphosphonium chloride (TPP; Sigma 675121), PQ dichloride (Sigma 36541), and NAC (Sigma A7250) were dissolved in ddH2O to make 500 mM stocks; FCCP (Sigma C2920), Antimycin A (Sigma A8674), Ionomycin (Sigma 407953) were dissolved in ethanol as 10 mM, 45 mM, and 10 mM stock solutions, respectively. For acute drug treatments, young adults were transferred to freshly made NGM drug plates 1–2 hours at room temperature before needle wounding. The synchronized young adults were transferred to freshly made NGM drug plate and then imaged using Zeiss Discovery LSM880 or the spinning disk confocal.

**Wounding, wound closure, and survival assay**. We wounded animals using femtosecond or Micropoint UV laser or with single stabs of a microinjection needle to the anterior or posterior body 24 h after the L4 stage. Actin ring and mitochondrial images were taken using LSM710 confocal microscope (63×, NA 1.4 objective) or spinning disk microscope (100×, NA 1.46 objective), Laser wounding images were taken using spinning disk confocal microscope (Andor 100×, NA 1.46 objective with IQ CORE image software). Actin ring quantitation and survival rate were performed as previously described[23]. Fragmented mitochondria were counted in a region of interest (ROI) 100 μm² within 20 μm to the wound site (4 ROI per animal) (Fig. 1d). The spread of mitochondrial fragmentation from the wound site was measured as the farthest distance of the fragmented mitochondria to the laser wounding site using ImageJ or MetaMorph software. Some metrics were normalized to WT = 1.

**GFP nanobody mediated protein degradation**. GFP nanobody tissue-specific knockdown was performed as follows. First we generated the strain fzo-1::GFP (zju136); Pcol-19-lifeactin::pHtomato (zjuSi22) to analyze actin ring diameter. Tissue-specific GFP nanobody degradation plasmids were generated using LR recombination. Plasmids for Pcol-19-vhh-ZIF1(zjuEx99), Psur-5-vhh-ZIF1 (zjuEx220), Pmyo-3-vhh-ZIF1(zjuEx154) were injected into N2 animals and the transgenic animals were then crossed with either Pcol-19-Lifeact-pHtomato or fzo-1::GFP; Pcol-19-lifeactin::pHtomato. Needle wounding and actin ring diameter measurements were performed as above.

**CRISPR-Cas9 mediated mutagenesis**. MIRO-1 mutagenesis was performed using the CRISPR-Cas9 system[58]. A mixture of plasmids containing pSX709 (*pU6-BseRI-BseRI-sgRNA-miro-1*) 50 ng/μl, and pSX524(*Peft-3-cas9-NLS-pU6-dpy-10 sgRNA*) 50 ng/μl was injected into the animals with a different transgenic or mutant background (see Supplementary Table 5). We screened for in-frame deletion mutations by DNA sequencing. All alleles were outcrossed before analysis. sgRNAs and deletion information are listed in Supplementary Tables 3 and 5.

**Heat shock**. P*hsp-16.2::fzo-1* transgenic animal heat shock experiment was performed as follows essentially described elsewhere[61]. Briefly, L1 worms were heat-shocked at 32 °C for 4 h, then incubated at 20 °C overnight before proceeding to needle wounding at the young adult stage.

**The single worm RNA sequencing**. Single worm RNA sequencing used a protocol modified from single-cell RNA sequencing for mouse cells[62]. A single young adult animal was transferred into 2 μl lysis buffer and lysed by grinding. In all, 1 μl of each oligo-dT primer (10 μM) and dNTP (10 mM) was added into the PCR tube and heated at 72°C for 3 mins then cooled for 2 min. 6 μl reverse transcription mixture (100 U SuperScript II reverse transcriptase (Takara), superscript II first-strand buffer, 1 U RNAase inhibitor (Vazyme),10 M betaine (Sigma), 6 mM MgCl₂ (Ambion), and 100 μM TSO primer) was then added directly and incubated using thermal cycle: 90 min at 42 °C, 15 min at 72 °C and hold at 4 °C. cDNA samples were amplified with 10 μl KAPA HiFi HotStart ReadyMix (Kapa Biosystems) and 12.5 μl 10 μM IS PCR primers. The purified cDNA was fragmented by TruePrep DNA Library Prep Kit V2 for Illumina (Vazyme Inc) Hiseq X 10 system sequencing. Primers are listed in Supplementary Table 3.

**RNA-sequencing analysis**. Each sample was analyzed in three biological replicates. Paired-end RNA-seq reads were generated on the Illumina HiSeq X10 platform. Clean reads were mapped to rRNAs and tRNAs firstly, and those unmapped reads were mapped to the *C. elegans* genome (ce11) with gene annotation WS258 using STAR[63] version 2.5.3a under default parameters except 'out Filter Match N min 40'. The Feature Counts program from R subreads package[64] was used to count reads mapped to each gene for all samples. The gene count by sample tables was used for differential expression analysis with DESeq2[65]. The cutoff for differential expressed genes (DEGs) were: Benjamini-Hochberg adjusted p-value less than 0.05 and foldchange larger than 1.5. GO-term analysis of DEGs was done with DAVID[66]. Figures were made with the ggplot2 package and Complex Heatmap package in R[67].

**Mitosox staining and imaging**. MitoSOX Red (Molecular Probes, M36008) staining was performed as follows. Briefly, young adult animals were wounded by needle and then transferred mitoSOX Red staining solution (5 μM) The animals were stained in the dark with gentle shaking for 20 min at room temperature. Stained worms were washed three times with M9 before imaging using a 561-nm excitation laser.

**RNA isolation and quantitative PCR (qPCR) analysis**. Total RNA was extracted from 30 young adult animals using TRIzol (Invitrogen, Carlsbad, CA, USA) and quantitated using a NanoDrop spectrophotometer (Thermo, USA). First-strand cDNA was synthesized by using the ReverTra Ace qPCR RT Kit (Toyobo, Japan). Primers for genes (Supplementary Table 3) were designed using Primer Premier 5 (Premier Biosoft). *act-1* was used as an internal control. The reaction mixtures were prepared according to the SYBR green kit instructions (Vazyme, China) and real-time quantitative reverse transcription PCR (RT-PCR) was performed using a LightCycler 480 II (Roche, Switzerland). The cycling protocol was: 95 °C for 2 min, followed by 40 cycles at 95 °C for 10 s, and 60 °C for 40 s. Relative expression levels were calculated using the $2^{-\Delta\Delta Ct}$ method.

**Cell culture and scratch assay**. U2OS cells (ATCC) were cultured in DMEM (GIBCO) supplemented with 10% FBS (Hyclone), 2 mM glutamine, 100U/100 μg/ml penicillin/streptomycin (GIBCO). The cells were seeded in a 4-well microplate and grown overnight to reach 90% confluence. To visualize mitochondrial morphology, cells were incubated in mitoTracker Green (Molecular Probes) and subjected to scratch wounding as described[68]. The mitochondrial morphology was taken using spinning disk confocal and images analyzed using MetaMorph. Each cell was classified as displaying tubular or fragmented mitochondria. The statistical details are in Source Data file.

**Visualization of MIRO-1**. GFP::*miro-1(zju21)* and *mKate2::miro-1(zju211)* knock-in strain were subjected to laser or needle wounding. Animals were mounted on 10% agarose pads in M9, in 12 mM levamisole and imaged before and after wounding using a Zeiss 880 confocal microscope. Green fluorescence was visualized with a 488-nm laser, and red fluorescence was visualized with a 561-nm laser. To score GFP::MIRO-1 on the membrane of mitochondria, fluorescence profiles at different sections of the mitochondria were obtained using the 'Line Scan' tool in MetaMorph. GFP::MIRO-1 was scored as localizing to the membrane if the peak of green intensity (GFP::MIRO-1) and the peak of red intensity (mitochondrial matrix mKate2) were not overlapping.

**Imaging and mitochondrial morphology analysis**. Images were taken using IQ image software (IQ CORE, Nikon) and analyzed using MetaMorph (Molecular Devices, San Jose, CA) and ImageJ (https://imagej.nih.gov/ij/). We scored mitochondria as fragmented if they were separate from other mitochondria and had a spherical or oval shape. The distance of fragmented mitochondrial to the wound site was defined as the farthest fragmented mitochondria to the wound site. We searched for fragmented mitochondria around the wound site and drew 20 lines for each wounded animal. The farthest distances were then averaged. All quantitation was performed by an observer blind to genotypes.

**Zebrafish tail-fin wounding and mitochondrial imaging**. WT zebrafish (AB line) larvae from the zebrafish center at ZJU. 3 days post-fertilization (3 dpf) larvae were cultured in E3 medium (5 mM NaCl, 0.17 mM KCl, 0.33 mM CaCl2, 0.33 mM MgSO4) containing 5 μM mitoTracker Green (Molecular Probes) for 2 h. Larvae were then washed three times before imaging. To examine the mitochondrial morphology, the larvae with mitoTracker Green staining were subjected to tail-fin tip wounding and then transferred to the imaging plate for confocal imaging[22]. No ethics approval was necessary for work with zebrafish larvae.

**Oxygen Consumption Rate (OCR) measurement**. OCR was measured using Seahorse Bioscience XF96[69]. Briefly, approximately 10-25 young adult worms were placed in each well and then measured by the XF96 respirometer. The experiment was repeated two times, with 8 wells for each genotype in each experiment. OCR per single worm was then normalized to the total number of animals.

**Statistical analysis**. All statistical analyses used Prism (GraphPad, CA). Two-way comparisons used a two-tailed unpaired *t*-test. One-way ANOVA for multiple comparisons, or the Fisher exact test for proportions.

**Reporting summary**. Further information on research design is available in the Nature Research Reporting Summary linked to this article.

## Data availability

The authors declare that all data supporting the findings of this study are available within this article, its supplementary information files, the peer-review file, the source data file, or are available from the corresponding author upon reasonable request. The RNA sequencing data generated and analyzed in this study are available upon request as well as from the Sequence Read Archive (SRA) at NCBI at the following accession code: PRJNA523321 [https://www.ncbi.nlm.nih.gov/sra/?term=PRJNA523321]. The source data underlying Figs. 1e, 2b, d, g, i, 3d, f, 4b, c, f, 5a, b, 6e–I, n–p, and 7b-e, and Supplementary Figs. 1d, f, 2c–e, 3c, 4f, g, 5a, g, and 7a, c, d, e, g–i are provided as a Source Data file.

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

## Acknowledgements

We thank C.L. Yang, Y. Jin, and W. Mair for reagents, L. Yu, G. Bai, Q. Sun, W. Liu, G. Ou for comments on the manuscript, and Xu lab members for support and discussion. We thank P. Xu and Z. Hong for help with zebrafish larvae and U2OS cell culture, and T. Zhuang for making strains. Some super-resolution images were taken using Zeiss 880 at the imaging center of the core facility at Zhejiang University School of Medicine. Several *C. elegans* strains were provided by the CGC, which is supported by the NIH Office of Research Infrastructure Programs (P40OD010440). This work is supported by the Ministry of Science and Technology of China (National Key Research and Development Program of China grant 2016YFC1100204), the National Natural Science Foundation of China (91754111, 31972891, 31671522), and 'One Thousand' Youth Talent Program to S.X., National Natural Science Foundation of China (31701271) and the China Postdoctoral Science Foundation (2018M630668) to J.Z., 'One Thousand' Youth Talent Program to Y.Z. Initial stages of this work were supported by NIH grant R01 GM054657 to A.D.C.

## Author contributions

S.X. conceived the study, designed the experiments, and performed initial mitochondrial fragmentation analysis as postdoctoral researcher in A.D.C.'s lab, with NIGMS support. S.X., H.Z., and Jz.Z. performed RNAi screen. H.F. made constructs, performed the wound closure, TMRE staining, Tissue-specific knockdown, RNAi, and fluorescence microscopy. H.Z. made constructs, performed CRISPR-Cas9 mutagenesis, knock-in, and wounding experiments. Jh.Z. performed single worm RNA sequencing, X.Y. and Y.Z. performed RNA sequencing data analysis. X.M. made transgene constructs and *fzo-1* GFP knock-in animals. J.X. performed qPCR experiments. H.F. and H.Z. performed microinjection experiments. S.X. performed cell scratch assay and zebrafish wounding experiment. S.X. and A.D.C. wrote the manuscript.

## Competing interests

The authors declare no competing interests.
