## [Peer Review File · Nature Communications]

REVIEWERS' COMMENTS:

Reviewer #2 (Remarks to the Author):

I do not have further comments, except that new Suppl. Fig 6E should more explicitly state whether mRNA or protein levels are measured as "Fold induction". The relevant measure to address my concern are protein levels of course, not mRNA. If the authors measured cyp-13a8 expression with their GFP line (Figure S6f), as I suspect, they should provide time-matched representative images of GFP-cyp13a8 around wound sites underneath the actin images in addition to their quantification.

Response to Reviewers' comments (our responses in blue):

Reviewer #2 (Remarks to the Author):

I do not have further comments, except that new Suppl. Fig 6E should more explicitly state whether mRNA or protein levels are measured as "Fold induction". The relevant measure to address my concern are protein levels of course, not mRNA. If the authors measured *cyp-13a8* expression with their GFP line (Figure S6f), as I suspect, they should provide time-matched representative images of GFP-*cyp13a8* around wound sites underneath the actin images in addition to their quantification.

We thank the reviewer for all the help on the manuscript and agree with the new revision.

Regarding the Supplementary Figure 6e, we detected the mRNA transcription of *cyp-13A8* by using qPCR method in a time-course manner and observed an increased expression level of *cyp-13A8* after wounding (measured as "Fold induction"). We now have revised the manuscript by clarifying that the *cyp-13A8* mRNA is measured, and have added this information in the text, figure, and figure legend. We certainly agree with the reviewer that protein expression is important, and our current data, together with Figure 6m-n are able to demonstrate that the upregulation of *cyp-13A8* is required for the actin polymerization-based wound closure. We hope that our clarification is sufficient.